# The combined effect of eye movements improve head centred local motion information during walking

**Szonya Durant**◉*, **Johannes M. Zanker**

Department of Psychology, University of London, Egham, England, United Kingdom

* szonya.durant@rhul.ac.uk

## Abstract

Eye movements play multiple roles in human behaviour—small stabilizing movements are important for keeping the image of the scene steady during locomotion, whilst large scanning movements search for relevant information. It has been proposed that eye movement induced retinal motion interferes with the estimation of self-motion based on optic flow. We investigated the effect of eye movements on retinal motion information during walking. Observers walked towards a target, wearing eye tracking glasses that simultaneously recorded the scene ahead and tracked the movements of both eyes. By realigning the frames of the recording from the scene ahead, relative to the centre of gaze, we could mimic the input received by the retina (retinocentric coordinates) and compare this to the input received by the scene camera (head centred coordinates). We asked which of these coordinate frames resulted in the least noisy motion information. Motion noise was calculated by finding the error in between the optic flow signal and a noise-free motion expansion pattern. We found that eye movements improved the optic flow information available, even when large diversions away from target were made.

## Introduction

Eye movements are most often investigated in the lab when a person is sat observing a computer screen; here they are typically rapidly scanning the scene [1]. Although in the modern world, this is increasingly an ecologically valid scenario, still a great deal of our lives is spent walking around. In this situation eye movements need to take on the additional role of helping to guide body movements and also stabilise the visual scene. Whilst walking, optic flow, the pattern of visual motion caused by our movement through the environment is considered by many to be an important cue to our own direction of heading (e.g. [2]). Small stabilising eye movements that keep the eyes focussed in the intended direction of heading, cancelling out head motion, by counter-rotating, should help to more clearly register the typical expansion optic flow pattern associated with self-motion. However, large saccades exploring the scene will cause spurious motion signals. A great deal of work has dealt with the problem of how to extract heading direction information from optic flow in the presence of eye movements by

**Data Availability Statement:** Recorded .avi movie clips and corresponding eye position data and description of which frames were sued in which conditions can be found on the external repository Zenodo database at https://zenodo.org/record/

545820#.WO5b6PkrK71 doi: 10.5281/zenodo.
545820.

**Funding:** The authors received no specific funding for this work.

**Competing interests:** The authors have declared that no competing interests exist.

setting up theoretical flow fields and models or using artificially recreated motion stimuli [3–5]. Some previous work has investigated the properties of the optic flow in scenes recorded from the real world in term of the statistical distributions of motion directions and magnitudes without eye movements [6,7]. To our knowledge no work so far has set out to assess the effect of eye movements on optic flow information in a real-world walking situation. We investigate to what extent eye movements pose a problem in terms of introducing noise into the signal available for heading extraction in this context. The examples above consider eye movements as a signal that needs to be detected ad corrected for, here we ask: what effect do they have if left uncorrected?

Several studies over the years have measured the eye movements during walking in terms of their role in stabilization, measuring the vestibular-ocular response that is triggered by the movement of the body. Walking has been measured on treadmills, where no optic flow is produced apart from the small amount of optic flow that helps regulate body posture [8,9]. In this case stabilising eye movements matching the phase of the head movements suggest an active role for gaze stabilisation. Descriptions of eye movement whilst walking ahead in space have also been studied in detail. Grasso et al. [10] considered the situation of people walking, asking them to head for a point round a bend, and found eye movements leading head and body movements, searching out the next direction point, in synergy with the body orientation required. Hollands et al. [11] found similar anticipatory eye movements when a new heading direction was cued part way through the walk trajectory, as did Bernardin et al. [12] for a curved trajectory reproduced from memory. Imai et al. [13] instructed participants to walk straight ahead for 3m, then to turn 90˚ and carry on straight for another 3m, asking participants to either look at the trajectory marked on the floor or to look straight ahead. They found that whilst orienting mechanisms dominated head, body and eye movements during the turn, during straight walking, it was image stabilization due to small corrective eye movements that was crucial in the coordination of these movements.

More recently Marius t'Hart and Einhäuser [14] and Matthis, Yates and Hayhoe [15] took these experiments out of the above laboratory settings to investigate eye movements and head movements in natural environments. Participants wore eye tracking glasses whilst navigating various real world terrains. Marius t'Hart and Einhäuser [14] used a terrain with irregular steps and cobbles, with no task restrictions on where to look. Matthis, Yates and Hayhoe [15] used a flat, medium and rough terrain, again with no instruction other than to walk from start to finish of a short route. Both articles found that the difficulty of the terrain greatly affected eye movement distribution. Eye movements are influenced by the task and in different terrains the implicit task of maintaining stability varies as different demands are made on regulating posture. In the work presented here we are interested in the case where the terrain does not provide difficulty, so that foot placement guidance is less necessary, so we can focus on the trade-off between stabilising and scanning the scene.

Pioneering work using the earliest versions of mobile eye tracking revealed fascinating aspects of eye movements during motor tasks and using real world environments [16]. This work demonstrated just how task-focused eye movements became in these situations and revealed the dynamics of the eye movements, jumping ahead to the next relevant part of the scene. Eye movements were also measure during walking, where participants were navigating through difficult terrains. Locomotion on foot was further investigated by Land and Tatler [17], in particular describing the two main possibilities for guiding our own motion as either by maintaining a target at a fixed egocentric direction, versus using the focus of expansion (FOE) in the motion pattern to steer our direction. When we are walking, the pattern of motion created by our own movement is that of expansion and the centre of expansion of this motion is usually congruent with our direction of heading. There is some debate as to whether

optic flow is necessary for guiding our locomotion—various papers suggest eye-rotations don't cause a problem because information about heading or path information is available that is independent of eye-movements [18,19]. Land & Tatler [17], referring to Warren et al. [2], conclude that when optic flow signals are available, they are used. Further models also make use of the geometric properties of optic flow to propose a model of locomotion guidance [20].

It is clear that eye movements play a role in stabilizing the image of the scene whilst walking, which may help improve the quality of the optic flow. We have also discussed the role of eye movements in scanning the scene [1]–in this case and also in the case of tracking moving objects they pose a problem for motion extraction [3,21] and some have proposed mechanisms that could solve this problem [22]. Here we ask: in natural walking conditions how much do eye movements distort the optic flow pattern on the retina? Do eye movements require correction as described above to extract optic flow?

In order to answer this question we consider the situation of simply viewing a full rich natural visual scene, whilst walking through it, and assess how typical exploratory eye movements affect the optic flow as measured from local motion on the retina. These movements comprise a combination of scanning and stabilising movements. Local motion is defined in terms of the motion vector available at each small patch in the image that has a direction and a magnitude. The approach taken here is to simply assess the amount noise in the low level motion information available to be processed later on by the brain in order to extract heading direction. We do not propose a model of how the brain achieves this, we simply evaluate the information available on the retina for such further processing. In order to achieve this, we use mobile eye tracking in which a forward pointing head mounted scene camera records movies in head-centred coordinates and at the same time the gaze is tracked to find the direction the person is looking in within the recorded scene. We use mobile eye tracking to compare the motion information in head-centred coordinates as provided by the scene camera, representing the situation as if the eyes were static, with the motion information available on the retina, by using the recorded eye movements to generate image sequences in retinocentric coordinates, i.e. the image sequence that would be available to the brain. A similar approach for estimating retinal motion was used in a recent conference presentation [23]. We compare results from a novel method developed here for a simple optic flow template matching procedure to estimate the amount of noise in the optic flow signal, where we consider any signal not informative to the extraction of focus of expansion as noise, based on the method used in Durant and Zanker [7]. This method estimated how much noise there was in the motion information when using it to estimate the focus of expansion of the optic flow. In that work it was found that even a stable forward moving camera produced very noisy estimates for the focus of expansion. We expect in our situation therefore that the head mounted camera will produce very noisy estimates (due to head motion) and we aim to investigate if the net result of eye movements is to reduce this noise. We measure this in two different environments. A target in the scene towards which the participants are heading along a straight trajectory allows us to extract a real world frame of reference for the eye positions. By establishing a measure for the 'quality' of motion information available for extracting our own direction of heading we aim to establish the effect of eye movements on optic flow.

## Materials and methods

### Pilot study

This study began as a smaller study with 4 participants, not all of whom took part in all conditions and an earlier version of the Tobii eye tracking glasses was used. In these data we found that participants very rarely looked away from the target, so in the full experiment instructions

were altered slightly to encourage them to behave more naturally. Some of the verification of the methods we used was carried out on these pilot data. Encouragingly the overall pattern of results between the pilot and full experimental runs remained the same.

## Participants

10 participants were recruited from the undergraduate students at Royal Holloway, University of London. They had normal or corrected to normal vision, and were naïve to the purpose of the study. The experiment was approved by the Royal Holloway Psychology Departmental ethics committee procedure.

## Recording

An indoor office corridor location and an outdoor woodland path was chosen (see Fig 1). A highly visible bulls-eye target was placed at each end of the corridor at around 1.5m height and inserted on a pole into the ground next to the outdoor path at around 1m height. Participants were instructed to walk in a straight line towards the target as if they were heading towards it to retrieve a book from a shelf, but to feel free to look around on the way as they wish. They repeated this each way up the corridor 3 times each. Then they were asked to do the same again, but this time keeping their eyes deliberately fixed on the target the best that they could. In the outdoors scene they first headed straight towards the target one way up the path, the first 3 times they were asked to walk naturally and the second three times to fixate and then this was repeated in the other direction down the path. In both scenes the walking surface was horizontal and smooth and the target was clearly visible ahead, with no obstacles, so participants could walk easily in a straight trajectory, with no need to look down for obstacles. We ensured there were no major distractions, such as people walking through the scene, directly in the field of view. The 'fixate' condition served partly as a check of our calibration and was important to see the limit of how well the eye tracker coordinates can be used to stabilise the images and produce the best possible motion information. It provides us with a baseline for the noise that is present in the signal under these optimal conditions for stabilisation, when most eye movement serves to improve signal and incorporates the noise in the gaze estimates. We refer to the other condition as the 'natural' walking condition from here on, although of course the participants knew they were being eye tracked and could interpret the task in many ways. Sometimes eye tracking failed or we were constrained by circumstance to not obtain 3

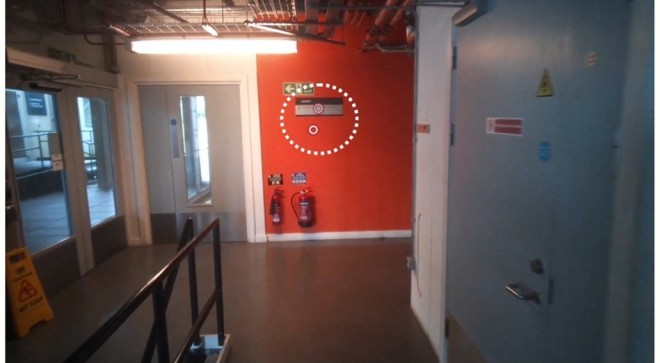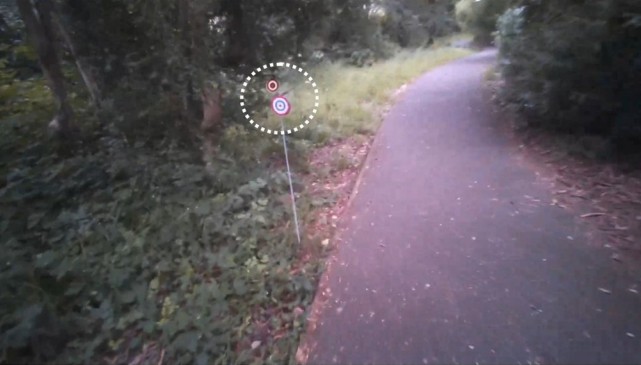

**Fig 1. An example frame recorded by the scene camera from the indoor office corridor and outdoor woodland path environment, with eye gaze position within the frame indicated by the red filled circle.** Within the dotted white circle (added here to clarify the figure), both the eye gaze position and blue-red-white bulls-eye target can be found.

repeats in each direction. For each participant for each scene and task condition the values calculated are based on the average of at least 3 trajectories (out of the 6 trajectories recorded over both directions for each condition).

Participants wore Tobii 2 eye tracking glasses. The scene camera recorded at 25 fps and both eyes were tracked at 50 fps to produce a single gaze point estimate. Tobii studio software interpolated gaze positions, returning a position within each frame of the recorded scene. The length of the walk was approximately 9m, which participants completed in about 10s, resulting in around 200–300 frames of recording per each walk. The scene camera field of view was 82˚ horizontally and the 1920x1080 pixel frames (we base our conversion between pixel and degree of visual angle on these values) were converted from .AVI files into image sequences. Distances in the scene camera frames are reported in pixels to avoid confusion later with the noise estimate that is reported in angles.

## Processing

In addition to the gaze position within each scene camera frame, we extracted the target position in each image by manually tagging the target centre to retrieve its image coordinates. For each clip we described the eye movements in terms of the standard deviation of the Euclidean distances of the gaze position from the target (i.e. how much the gaze positions varied relative to the target), calculated on each frame, see Eq 1.

$$\text{Variation in relative gaze position} = s.d.(abs(\sqrt{(target_x - gaze_x)^2 + (target_y - gaze_y)^2})) \quad \text{Eq 1}$$

Where $target_x$ is the x coordinate of the target in a scene camera frame and $gaze_x$ is the x coordinate of the gaze point in a scene camera frame. The distances are calculated for each frame in a clip and the standard deviation of these differences forms the metric for the variation in gaze position for a given clip.

We also classified gaze position as 'on target' or 'away from target' (i.e. making large excursions away). To do this, we first divided the eye movement sequence into intervals between rapid changes in position (more than a 100 pixels (~4 degrees) in one frame–from the raw output frames). This parameter describes a very simple saccade detector, tailored to our data. Thus we divided the gaze trajectory into continuous parts. Over each of these intervals we calculated an average distance away from target. We assumed that the smallest distance observed in a complete walking episode that lasted over 10 frames would represent an interval in which the gaze was tracking the target. The average distance between target and position over this interval was defined as the baseline distance (instead of from the average over the whole sequence as this was affected by large distances away from target and did not result in effective classification of on-target and divergent gaze positions). We calculated the distance of the gaze position from baseline on each frame (Eq 2).

$$distance\ on\ each\ frame = abs(\sqrt{(target_x - eye_x)^2 + (target_y - eye_y)^2} - baseline). \quad \text{Eq 2}$$

Time intervals with an average distance more than 100 pixels (~4 degrees) in distance from the baseline distance in the raw output frames were labelled as 'away from target'. In Fig 2 we illustrate the results of this process for an example set of x and y coordinates over an image sequence. We can see that the gaze in this example follows the target a large proportion of the time and that large movements away are classified as 'away from target'. Upon visual inspection of all the sequences this was mostly successful in classifying the gaze positions. In some sequences the gaze does not appear to be tracking the target often, although in all sequences we found the minimum of required 10 continuous frames on target. As a check, only including

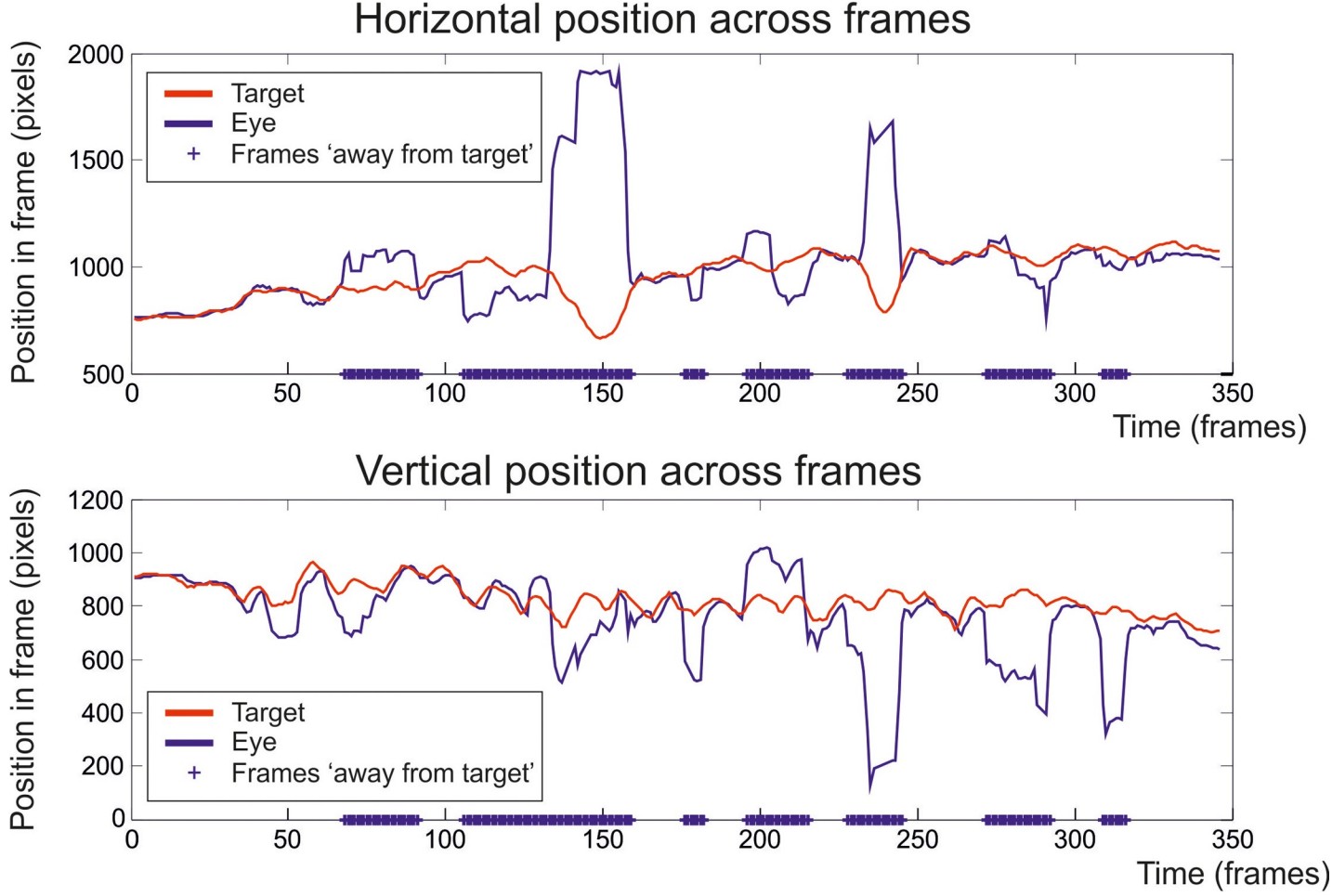

**Fig 2. An example showing gaze positions divided into 'on target' and 'off target'.**

distances from the (raw output) frames classed as 'on target' reduced the average distance error from baseline from 106 pixels (averaged over all the frames) to 32 pixels (averaged over only the 'on target' frames) and the standard deviation from 103 pixels (s.d. over all the frames) to 30 pixels (s.d. over only the 'on target' frames), suggesting our method for measuring excursions away from target was effective. See Fig 2 for an example.

To compare motion information in retinocentric versus head-centred sequences (Fig 3) we first realigned each frame from the scene camera image sequence so that the gaze position was at the centre of the frame, shifting the image in the $x$ and $y$ directions. If the gaze information was lost on a particular frame, the previous frame was repeated–this provides no additional motion information. On average 3.8% of the frames were interpolated. We then added a circular Gaussian window to all images in both the head-centred and retinocentric image sequences to avoid the effects of edges appearing in the retinocentric images and to have equal areas of the visual scene in the two types of sequences to be compared. The recorded images were reduced to half the resolution (but keeping the FoV) by down sampling and cropped in the horizontal dimension to the width of the circular window, which had the diameter of the vertical FoV of 46 degrees, so 540x540 pixel images were the input for the model (Fig 3). There was no difference in the mean luminance level of two types of image sequences, it was the same at 0.48, and the s.d. was the same at 0.01 (on a scale of 0–1 possible image values) for both.

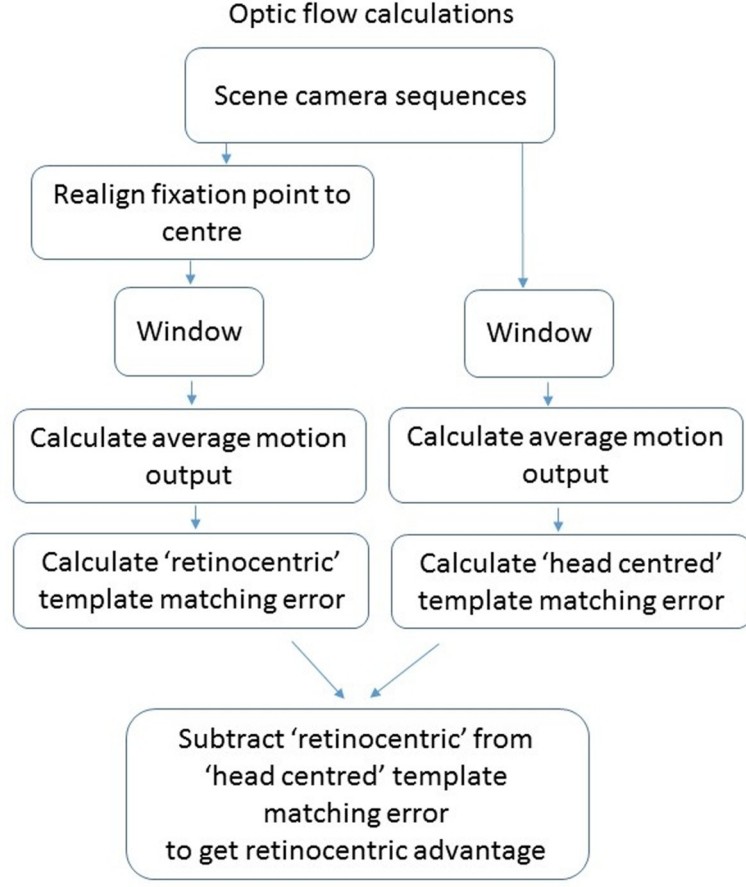

**Fig 3. Diagram illustrating steps for processing optic both from 'head centred' and 'retinocentric' coordinates in order to calculate template matching error.**

The 2DMD local motion model, based on spatio-temporal correlation was used, as has been applied previously to calculate local image motion in natural scenes [7,24]. The 2DMD model is spatio-temporal correlation model that correlates frames over space and time. The spatial sampling base and temporal sampling base determine the distance and time between the image patches to be compared. We used a spatial sampling base of 8 pixels (in input frames for the model about 0.6˚ of visual angle) and a temporal sampling base of 10 frames (about 400ms). This was chosen on the basis of piloting to find parameters where the motion model gave measurable optic flow responses. With a smaller temporal sampling base, the motion output became very noisy, and we did not want a longer temporal sampling base, 400ms seems reasonable to integrate optic flow, as MST neurons can take around 400ms to reach peak, so this can be taken as their temporal integration time [25,26] although this is a fairly long integration time. The sampling rate of the scene camera means that high speed components of the visual scene due to eye movements will be lost. Instead we are sampling slow changes in input frames caused by the tracking of the target compensating for walking gait and by self motion. See the Discussion further on these issues. The relatively fine spatial sampling (comparison over small distances) is based on the high spatial frequency content of the scenes. Past work has shown that the spatial sampling base does not have an effect on the overall distribution of motion directions [7].

For each input frame the 2DMD model produces two output frames, one containing horizontal motion values, the other vertical, from these, motion direction and magnitude can be calculated (Fig 3). For each clip we can calculate the average motion pattern (averaged over all frames) available over the clip. We apply a simple template matching algorithm based on Durant and Zanker [7] to assess how much noise is in the clip compared to a pure motion expansion pattern. This method finds the image location that produces the best match point as a hypothetical focus of expansion (hFOE) and calculates the difference between the motion directions radially around the hFOE and the ideal radial expansion motion pattern around the FOE as illustrated in Fig 4A. The calculated difference at this point is the error value in degrees of mismatch between the motion distributions, i.e. the amount of motion noise with regard to extracting FOE, using template matching.

To calculate this error over a whole sequence we took the motion outputs from the 7th frame from the start of a walking sequence to the 5th frame before the end of a walking sequence, averaged the horizontal and vertical motion outputs separately to produce an average motion output for each and then performed the template matching calculation using this averaged motion output (excluding a central area of 52x52 pixels (in the input frames) to avoid the fixation marker in the image sequences produced by the eye tracking software interfering with template matching in the retinocentric images) (Fig 3). We do not suggest that the visual system integrates over 10s worth of information; we simply calculate a measure of motion information available to judge heading direction over a whole sequence. This averaging was necessary due to the noise in the local motion calculations in the scenes, as has been found previously [7]. In order to reduce processing time we found the minimum by calculating these error values in a simple grid search function. In Fig 4 we show an example of the averaged motion output (a) from a test sequence, the template matching error at every single location in the output (b), and how this corresponded to the target positions over the sequence (c).

## Results

### Indirect estimate of head movement

The variance of the target position in the scene camera can be used as a proxy for the amount of head movement in the sense that it is a measure of the displacement of the scene camera image due to head movement (see Table 1. for how much target position varied horizontally and vertically within sequences and across conditions). A 2x2 ANOVA with target position variance in the horizontal direction and also another 2X2 ANOVA with target position variance in the vertical direction as a dependent variable revealed that the target moved around more in the scene camera frames when the participants were walking naturally and also in the woodland scene (horizontal positions: task $F_{1,9}$ = 21.5, p<0.001, scene $F_{1,9}$ = 6.7, p<0.05; vertical positions: task $F_{1,9}$ = 30.5, p<0.0001, scene $F_{1,9}$ = 16.0, p<0.005,). This suggests the need to fixate vs the freedom to look around altered head movements and there was more head movement in the woodland scene than in the office scene.

### Eye movement statistics

We extracted the average variation in gaze position relative to the target, as a measure of how closely the target was tracked, across conditions. We tested how this varied by applying a 2x2 ANOVA (task and scene). We found a significant effect of task, i.e. participants adhered to instructions and tracked the target more closely in the 'fixate' condition (mean variation when fixating = 47 pixels, mean variation when not fixating = 163 pixels, in the raw input frames). We see that in the 'natural' condition there is large variation in the relative position of gaze to target, larger than the variation of the target position in the scene camera, showing that gaze

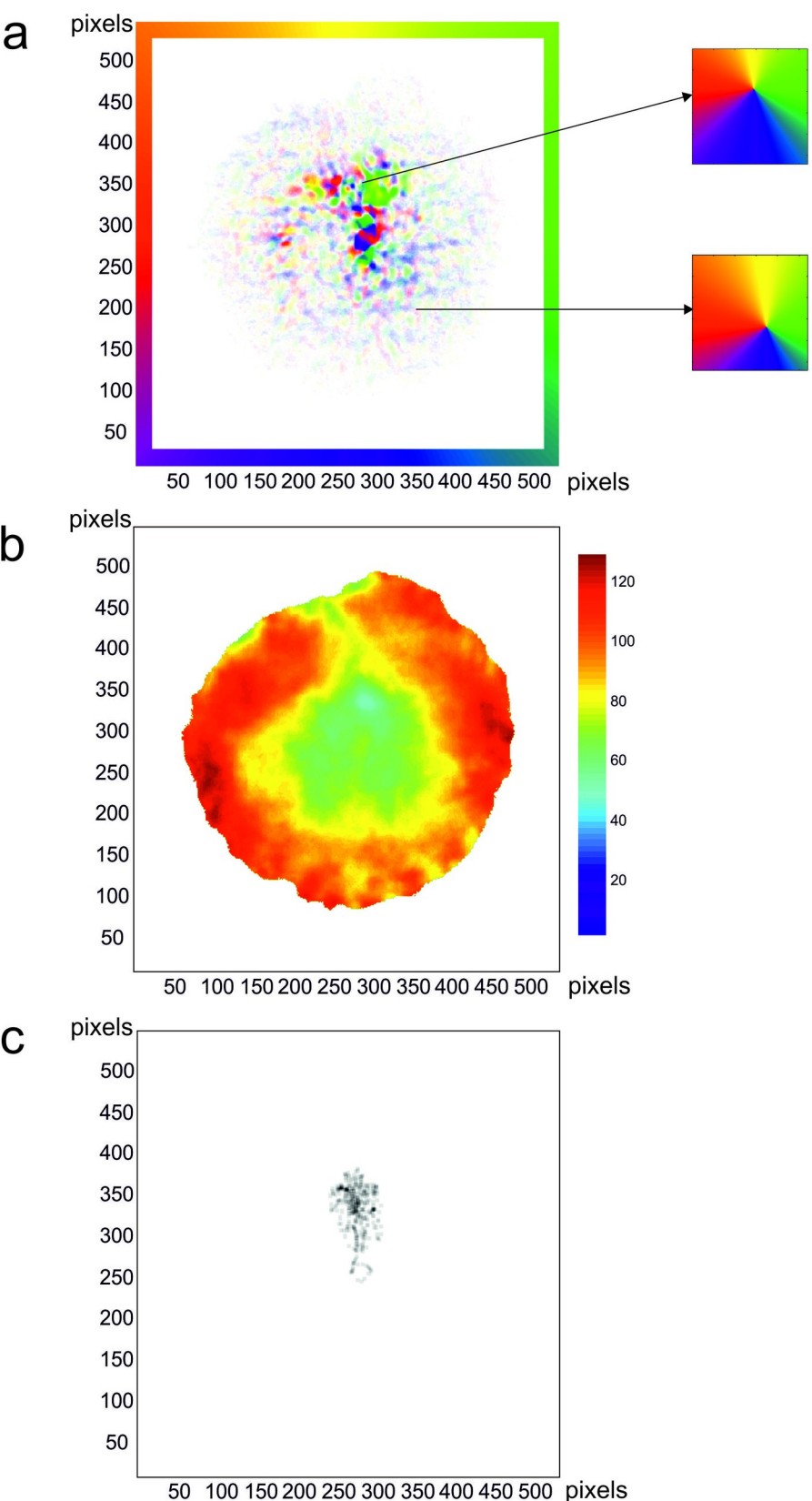

**Fig 4. All images shown in head-centred coordinates.** (a) Colour coded direction output (border indicates direction colour code) for a walking sequence recorded by the screen camera, average of 355 frames. Example templates that are applied at the shown points, from which matching error is calculated in (b). (b) Template matching error to the ideal expansion pattern scaled to min blue = 0˚ error, max dark red = 126˚ (i.e. there are some points where the angles are opposite to the template). Minimum value error = 49˚. White areas did not have enough valid motion values at each radial angle to be able to calculate template matching error. (c) The locations of the target from each of the head-centred frames over the sequence overlaid in a single image.

often did leave the target to scan the visual scene. We did not find an effect of scene, so although there was more head movement in the woodland scene, this did not affect how closely the gaze tracked the target.

We also extracted a measure of correlation over a whole sequence between the horizontal and vertical gaze positions and target positions. High correlations could be achieved in the fixate condition on average of 0.75, and in the natural condition a reduced correlation of 0.37 was found in both the x and y positions. The offset between time series that produces the highest correlation can be interpreted as the lag between the two signals, the average lag over sequences with a correlation above 0.75 was 1 frame in *x* and *y* coordinates, so effectively the eye was in synchrony with the target (we do not have accurate enough timing to infer anything from the 1 frame lag). This close match without a measurable lag demonstrates the close coupling between target and gaze position that has been suggested to be achieved by extremely fast subcortical vestibular connections [27].

The eye movements further reveal the difference between deliberately fixating on the target and moving more forward freely observing the scene, on average in the fixating the target conditions 95% of the time the gaze was classed as tracking the target (so people were performing the task as asked, with occasional involuntary movements away), whilst in the natural condition only 55% of the time. We conducted a 2x2 ANOVA on the proportion of frames on target to investigate any possible differences across task and scene. The mean proportion on target when the task was to fixate the target were: office corridor 0.96 (s.d. = 0.07), woodland path 0.93 (s.d. = 0.11) and in the freely walking condition: 0.54 (s.d. = 0.20), woodland path 0.52 (s.d. = 0.18). As expected there was a main effect of task ($F_{1,9}$ = 84.6, p<0.0001), but the type of scene had no effect on the amount of time spent tracking the target ($F_{1,9}$ = 0.64, p = 0.44), suggesting there was no extra need to look at the path in the woodland scenes or additional distractions.

### Comparison of template matching error between head and retina centred coordinates

The graph in Fig 5 shows the means and *s.d.* over participants for each of the conditions for the template matching error (see Methods and Fig 3). This measure calculates how noisy the

**Table 1. For each sequence and participant (N = 10), the s.d. of target positions in the scene camera raw output frames is calculated as proxy for the amount of image motion due head movement.** These are averaged for all the sequences recorded for one participant to produce one value. The value for the participant with the minimum and maximum s.d. of target displacement within the frames and the average values over the participants are shown.

| Units in pixels (images are 1920 x 1080) | | N | Minimum | Maximum | Mean |
|---|---|---|---|---|---|
| Horizontal | Office natural | 10 | 23.39 | 216.13 | 82.12 |
| | Office fixate | 10 | 16.58 | 56.55 | 31.54 |
| | Woodland natural | 10 | 32.32 | 181.31 | 118.72 |
| | Woodland fixate | 10 | 25.71 | 140.96 | 67.46 |
| Vertical | Vertical Office natural | 10 | 39.52 | 122.18 | 78.21 |
| | Office fixate | 10 | 24.75 | 62.53 | 40.30 |
| | Woodland natural | 10 | 57.45 | 121.05 | 94.82 |
| | Woodland fixate | 10 | 41.60 | 117.99 | 66.30 |

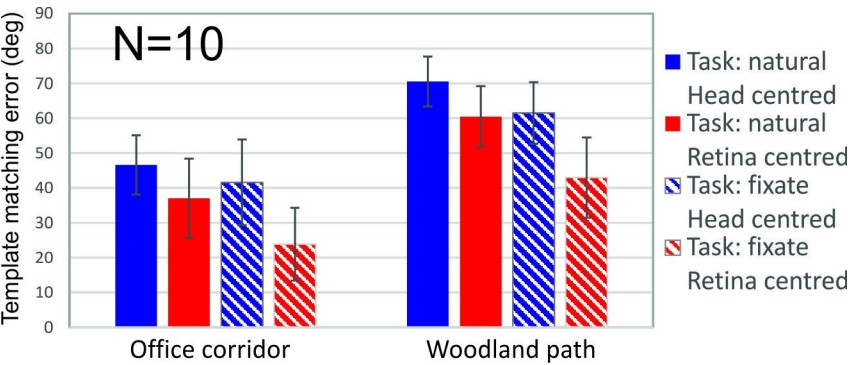

**Fig 5. The average template matching error for each condition.** Error bars are standard deviations calculated over participants' scores, but do not reflect significance as we used a within subject analysis (see Results section). In particular, the reduced error from head-centred (blue) to retina-centred (red) coordinates for the 'natural' task is significant. Template matching errors of 90 degrees would indicate a random motion pattern (and therefore no match).

overall signal over a sequence is in terms of how well it can be matched to the ideal expansion template. A perfect match would yield 0 deg (perfect match), a completely opposite template (contraction) would yield a 180 deg error (perfect anti-match), whereas a random motion pattern would yield a 90 deg error (random matches at each point varying between 0 deg and 180 deg) (this was verified). Note from the graph that the camera sequences are very noisy, and this is similar to what was found in Durant and Zanker [7]. We incorporated all the factors in a repeated 2 x 2 x 2 ANOVA, with the two scenes, the two task conditions (natural or fixed) and whether the motion information was in head or retinocentric coordinates. The dependent variable was the calculated template matching error (for each participant this was the average over all sequences in that condition). Most importantly in our main comparison of interest, which looks at the combined effect of eye movements versus no eye movements, there was a significant main effect of coordinate frame used, so there was less error in the retina-centred coordinates ($F_{1,9} = 35.0$, $p<0.0001$). Calculating the advantage for each sequence, the average improvement in retinocentric coordinates is 19 degrees. This shows that overall we can take the gaze coordinates from the eye tracker and get a more stabilised image. There was also a significant interaction between coordinates used and task condition, so the advantage of the retina-centric coordinates was greater when the task required fixation ($F_{1,9} = 19.3$, $p<0.0001$), suggesting that when the gaze tracked the target deliberately, this improved motion information quality. However, we also analysed the results separately for just the natural task condition and still found a significant effect of retinocentric vs head-centred coordinates. The average reduction of the template matching error in retinocentric coordinates in just the natural condition was 10 degrees, $F_{1,9} = 19.0$, $p<0.005$ (and there was also less template matching error in the office scene, $F_{1,9} = 39.8$, $p<0.001$, but no interaction, $F_{1,9} = 0.01$, $p = 0.91$). It is important to note that average retinocentric error was significantly different from chance (mean = 48.7 deg error, s.d. = 15.6 deg error) i.e. although the sequences are noisy, the reduction in noise from head centred to retinocentric coordinates is meaningful. Despite the large variability in gaze position relative to target in the 'natural' condition (see Table 1), there is still an improvement overall introduced by eye movements to the quality of the motion information.

We also found a significant main effect of scene, with smaller error in the office surroundings ($F_{1,9} = 41.2$, $p<0.0001$). This result, suggesting better quality of motion information in indoors environments, has been found previously [7]. We also found a significant main effect of task condition, so there was less error when people kept their eyes fixed in both coordinates

$(F_{1,9} = 40.3, p<0.0001)$, showing that eye movements other than tracking the target did increase error.

## Discussion

Whilst walking the rotary components of the head frame of reference are not all zero, and thus the optic velocity field will be the vector sum of the radial optic velocity field due to the translatory movement of the head frame of reference, and a rotary optic velocity field due to the rotation of the head frame of reference. Thus the optic velocity field affords noisy information about the direction of heading. The visual system could in principle deal with this problem by counter-rotating the eye in the head to cancel out the rotary movement of the head frame of reference. However, in addition to these counter rotations the eyes are also scanning the visual field for relevant information, which may hinder the ability to extract information about the direction of heading from the velocity field. Whilst mobile eye tracking offers the opportunity to study the interaction of body, head and eye movements in navigation there are considerable restrictions of the visual field as we elaborate below and only a rather slow and unreliable rotation signal of head movement is provided by the eye tracker gyroscope and accelerometer.

Given these theoretical and practical challenges, our approach is to restrict ourselves to a very confined question–the effect of gaze position during the approach of a target on the overall quality of optic flow information picked up by the retina from the environment, as compared with the scene camera information, where gaze position is not taken into account. This is a partial step to understanding the comprehensive processing mechanisms of optic flow in human agents moving through a natural environment.

In this study, in a natural walking situation, we were able to extract direction of heading over a sequence of frames from local motion by matching to an ideal optic flow template significantly better than would be expected with random motion information, showing that without further processing (i.e. computational steps for extracting complex motion patterns and reducing noise) there is a meaningful motion signal present at this stage. Moreover, we found that in these real-world examples of locomotion, the motion calculated in retinocentric coordinates—shifting the sequence frames to represent location relative to the direction of gaze, provided a better quality signal for the extraction of the direction of heading than the motion calculated in head centric coordinates, as recorded by the head-mounted camera. Eye movements reduced the motion template matching errors despite gaze excursions away from the target (as shown by the large increase in the difference in eye position and target position in the natural condition) that the participant was walking towards. The benefits for retinocentric coordinates were there despite participant being encouraged to freely look around naturally whilst walking.

Many different motion processing models could have been used, but as in this work we are only taking into account local responses to luminance based motion, the choice of model should not make much difference as most low-level motion models produce similar local outputs with stimuli containing broadband luminance-defined motion, as has been suggested by work that has used different model outputs to local motion [28,29]. Our simple local motion model required averaging over a large amount of time to be able to extract optic flow reflecting the self-motion of the participant. This is partly due to sparse nature of local motion signals available in real-life scenes [24] and may be due to the simplicity of our template matching procedure. The temporal scales used in our model may have reduced the effects of fast saccade type eye movements more than slower pursuit-like stabilising eye movements, suggesting that at the temporal scale need to extract informative optic flow information, large scanning eye movements contribute less to the extracted motion signal.

The visual system almost certainly relies on more sophisticated processes for extracting our self-motion, for example neural networks such as suggested by Lappe and Rauschecker [30] for detecting flow components. However, as a basic measure of information available, with few assumptions, we believe that our data suggest that in natural scenes self-motion could be extracted in such a way as to not be affected by the typical large gaze position digressions occasionally observed from the heading direction. It is not the aim of this article to discuss the best models of human motion perception, but merely to use a local motion based metric for evaluating the amount of motion information useful for extracting heading direction. Equally it is worth mentioning that although we here demonstrate that heading could potentially be recovered from the retinal flow field alone, extra-retinal information is also available to allow compensation for any eye rotations that might occur (e.g. [31]).

One limitation of our analysis is the reliance on a patch of information of around only 46 degrees of visual angle across. This is due partly to the limitations of having to match the area around the gaze position to the equivalent area in head-centred coordinates, when the participant is gazing close to the edge of the scene camera view, and this in turn is limited by the field of view of the scene camera. A large amount of research suggests that more peripheral motion cues are important for detecting optic flow, suggested by the very large receptive fields found in MST, sensitive to redial flow patterns [32] and corresponding psychophysical evidence for integration of radial flow over visual areas of up to 70 degrees [33]. However, some research also suggests that central vision is very important for using optic flow for heading estimation [34]. In our case this limited part of the scene contained enough information for successful extraction of the FOE, in that the motion information yielded smaller template matching errors than random motion would. However, our method does not allow us to estimate the effect of eye movements on more peripheral motion signals, which may be different. If only considering motion information as recorded by the scene camera, more peripheral motion components can be extracted in future work, as we will not need to window the scene camera frames as we did here.

In theory eye movements disrupt the optic flow based on what would be expected from self-motion, and this results in a complex computational problem [3,21]. Sophisticated models have been developed in the past for disentangling eye movement generated optic flow from that generated by our own motion, and indeed in lab simulations of the combined flow generated it is found that humans are able to disambiguate successfully [21]. Here, we have added to previous work describing the pattern of gaze position during navigation and describing mechanisms for extracting optic flow in the presence of eye motion by recording real optic flow and eye movements rather than using simulated versions. This work suggested to us that on the time scales we need to have enough motion information to extract the focus of expansion, eye movements away from the target we are headed towards, before moving back to target, did not appear to greatly affect the motion calculations needed to be able to use motion to deduce our direction of heading. Not surprisingly, when given the option, human walkers tend to look where they are going a large proportion of the time [20]. We would suggest that it is possible that the amount of time that walkers can afford to look away from where they are headed is limited by the need to not degrade the motion information and in the settings here they were able to keep within this limit.

The measures we used involved averaging over the whole walking sequence, of around 10s, but we are not suggesting that this is how the visual system extract heading from optic flow, it is merely a tool we use to assess the information available in the local motion. In fact, given the long time needed to extract a useful optic flow pattern, it seems likely that the visual systems carries this out in further steps that use the local motion field as input, for example as suggested

by Krapp and Hengstenberg [35]. However, this further processing will be reliant on the quality of local motion information, which is what we aimed to assess here.

Our results suggest that in the case of heading towards a fixed target the theoretical problem of disentangling rapid orienting eye movement induced retinal motion may not need solving. This may be due to the slow integration time used in our local motion model. When we tried a reduced integration time for extracting motion information, this resulted in meaningless motion information, with no ability to retrieve optic flow from camera or retinocentric coordinates. It may be that the effect of disruptive eye movements may be greater at a higher sampling rate, we show here that at a given time scale involving slow integration times, they do not disrupt motion information.

It must be noted that over the walk trajectories reported here, the distance of the target varied and it has been found that the types of eye movements used for stabilisation vary as a function of the distance of the target [8]. We remain agnostic as to the exact mechanisms causing the compensatory eye movements that produce stabilization whilst tracking the target and merely evaluate their overall effect. It is the accuracy of these movements that is responsible for how good the optic flow information is for extracting heading.

Whilst this article concerns itself with the use of optic flow for guided walking it is worth noting that there are other uses of optic flow such as postural control and flow-parsing. Future research could consider how eye movements may affect the motion signals available for these different tasks.

In our study the walking terrain was smooth and flat to allow examination of the trade-off between stabilizing eye movements and scene scanning eye movements, however adding uneven terrain and obstacles dramatically changes the pattern of eye movements [15,17] and in further work it would be interesting to see how these affect optic flow. Also, in our task participants merely needed to head forward along a straight trajectory, in the case of moving round corners past work has found that many large re-orienting eye movements are required [10,13] and it may be that more frequent, temporally adjacent eye movements may disrupt the self-motion induced optic flow, but this is a different case, where the heading target position is constantly changing, posing additional difficulties for models estimating the focus of expansion location. In our procedure there was no requirement to make scanning movements around the scene, although participants occasionally did so naturally. Adding additional tasks in a dual task walking paradigm may again induce more eye movements with a larger combined effect on the quality of optic flow available for the extraction of the focus of expansion. Adding a dual task (e.g. ticking boxes on a piece of paper or doing up buttons) has been shown to have effects on locomotion performance [36,37], suggesting motion information may be compromised. In a study where the additional task was to avoid other pedestrians, many saccades to the expected location of the appearance of a pedestrian were recorded [38], which may affect the quality of the motion information and also temporal requirements for control processes. In future work we aim to introduce such secondary tasks to measure how they affect the retinal motion information.

We have shown here that although self-heading direction can be retrieved from a head mounted camera, the motion correction carried out by the eyes does improve this, despite the dual role of eye movements as motion stabilisers and locators of relevant scene information. Deliberately keeping the eyes steady did improve the motion information, however retinal advantage is retained overall on average despite large saccadic eye movements, suggesting they may not need to be individually corrected for in order to successfully retrieve heading direction from optic flow.

## Acknowledgments

Thanks to Helen Scott for help with data processing, Tim Holmes and Scott Hodgins for help with loan of equipment and help with data collection and processing.

## Author Contributions

**Conceptualization:** Szonya Durant, Johannes M. Zanker.

**Data curation:** Szonya Durant.

**Formal analysis:** Szonya Durant.

**Investigation:** Szonya Durant, Johannes M. Zanker.

**Methodology:** Szonya Durant, Johannes M. Zanker.

**Project administration:** Szonya Durant.

**Resources:** Szonya Durant, Johannes M. Zanker.

**Software:** Szonya Durant, Johannes M. Zanker.

**Visualization:** Szonya Durant.

**Writing – original draft:** Szonya Durant.

**Writing – review & editing:** Szonya Durant, Johannes M. Zanker.

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
