## [Decision Letter · Decision Letter 0]

8 Nov 2019

PONE-D-19-26713

Local motion information is improved by eye movements during walking

PLOS ONE

Dear Dr Durant,

Thank you for submitting your manuscript to PLOS ONE. After careful consideration, we feel that it has merit but does not fully meet PLOS ONE’s publication criteria as it currently stands. Therefore, we invite you to submit a revised version of the manuscript that addresses the points raised during the review process.

In general, both reviewers seems to be asking for a more nuanced description of the rationale and the limitations inherent in the study toward realizing the stated objectives. Reviewer number 1 has very specific queries that could be addressed. Reviewer 2 has a more generalized critique that merits consideration. I am confident that this could be addressed in the discussion with a section addressing methodological and theoretical limitations. 

We would appreciate receiving your revised manuscript by Dec 23 2019 11:59PM. To enhance the reproducibility of your results, we recommend that if applicable you deposit your laboratory protocols in protocols.io, where a protocol can be assigned its own identifier (DOI) such that it can be cited independently in the future. For instructions see: http://journals.plos.org/plosone/s/submission-guidelines#loc-laboratory-protocols

We look forward to receiving your revised manuscript.

Kind regards,

Doug Wylie, Ph.D.

Academic Editor

PLOS ONE

Journal Requirements:

Reviewers' comments:

Reviewer's Responses to Questions

**Comments to the Author**

1. Is the manuscript technically sound, and do the data support the conclusions?

Reviewer #1: Yes

Reviewer #2: Partly

2. Has the statistical analysis been performed appropriately and rigorously? 

Reviewer #1: Yes

Reviewer #2: I Don't Know

3. Have the authors made all data underlying the findings in their manuscript fully available?

Reviewer #1: Yes

Reviewer #2: No

4. Is the manuscript presented in an intelligible fashion and written in standard English?

Reviewer #1: Yes

Reviewer #2: Yes

5. Review Comments to the Author

Reviewer #1: This manuscript describes a study that primarily sets out to determine whether extraction of motion cues from a moving scene can be improved by taking into account eye-movements. Because eye-movements serve a number of functions it is somewhat hard to determine whether the focus of the study is to examine stabilising eye-movements, scanning eye-movements or a combination of the two. Whilst I like the general idea (to look at eye-movements during real-world walking to determine how they might be used to help extract motion information) my sense after reading the paper is that the results mainly confirm what is already known: that some eye-movements have a useful role stabilising the retinal image when walking, but other eye-movements make it harder to extract motion signals from the retinal image compared to pure target fixation.

The approach in the paper is to ask participants to walk to a target whilst wearing an eye-tacker, then extract motion signals from the head-mounted scene camera video, and compare this to when motion-signal extraction uses an eye-movement transformation. Contained within the eye-movement transformations are a combination of stabilising and scanning components. The authors manipulate these components by having two different fixation conditions: ‘Fixate’ (where participants walked whilst smoothly tracking/fixating a target, so this predominantly contains stabilising eye-movements) and ‘Natural’ (where participants walked and looked to the target, but were encouraged to look around freely, so this presumably had more scanning eye-movements whilst also containing stabilising eye-movements). The authors demonstrate that they were best able to extract direction of motion information from scene camera images using stabilising eye-movements, but that ‘Natural’ eye-movements were better than no eye-movement signal. Because the characteristics of ‘Natural’ fixations seem to be unquantified it is hard to see how the authors can achieve their main goals: examining the “trade-off between stabilising and scanning the scene” (line 79-80), determining ‘how much do eye-movements distort the optic flow pattern on the retina?’ (line 102) and aiming ‘to investigate if the net result of eye movements is to reduce … noise’ (line 126). Each of these goals would seem to require a more detailed examination of what eye-movements were being generated in 'free' conditions (and quantifying them) during their walking tasks in order to then determine their contribution to the retinal patterns produced.

Other major queries:

Q1. There needs to be a fuller up-front rationale/explanation for the visual field sampling. The manuscript does mention this point in the discussion (line 371) but merely states that this is a limitation without giving the reader a sense of how this might impact upon generalisation of the findings. When reading the manuscript I had the following questions: There is a difference between the field of view of the camera and the actual visual field of participant - the 82deg horizontal camera view is much less than the visual field of the observer, leading to reduced peripheral signals. Because optic flow motion vectors are usually larger in the periphery could this not have led to underestimation of the impact of eye-movements on detecting self-motion? Linking this to line 198/199 the MS mentions ensuring there are ‘equal areas of the visual scene’ and then line 201-202 ‘cropped in horizontal dimension’ to ‘540x540 pixels’ – what field of view does this effectively reduce the visual field down to? My calculations based on line 213 (8 pixels = 0.3deg) would suggest this is 20.25 x 20.25deg? (in the discussion it mentions around 23 deg) I think it would be good to mark this effective size on Figure 1 to give the reader a sense of the areas of the scene actually contributing to the current analysis since currently there is a slightly misleading sense that the whole scene image is contributing (whereas much of the peripheral information won’t be I think).

Q2. Lines 202-204. Does stating the average luminance values (~0.5) on a scale of 0-1 of possible image values have any useful meaning? Presumably the outdoors scene was actually much brighter than indoors, but the camera recorded images of similar luminance across scenes due to reduced aperture size (or other scaling methods to prevent overexposure and thereby keep luminance values broadly similar)? Is that the point of the statement – to show that the recorded scene images were similar? If so providing average luminance values for both scenes might be necessary? I suggest giving the reader an idea of why this information is presented to aid interpretation.

Q3. I think there needs to be greater consideration of the particular types of eye-movements (i.e. saccades) that would be affected by the analysis/processing methods. Line 220 suggests that high speed changes are filtered out. This issue is somewhat acknowledged in the discussion but I am still unclear whether pooling across 10 frames (~400ms) removed all saccades but left other eye-movements intact, or whether other rapid eye-movements (e.g. OKN) might also be removed in this pooling. How might this affect conclusions about the role of eye-movements given that saccades will predominantly be supporting scanning behaviours, whereas stability would be supported by slower/lower frequency tracking eye-movements.

Q4. Template matching errors in the region of 25-70 degrees are reported but I initially struggled with interpreting these values. Lines 304-307 are very helpful in clearly explaining what the values mean. I would recommend adding to the Figure 4 caption an explanation that matching errors of 90 degrees would indicate a random motion pattern (and therefore no match).

Q5. Methods (Line 146-166). The authors need to clarify what sorts of trajectories were taken when walking. Given the images in Figure 1 and the description of the corridor I presumed that straight line walking was produced in both environments? (there are hints in various places this might be the case). However the outdoor scene seems to have the target placed off to the side of the path (figure 1). Did participants walk straight along the path or straight towards the target? Was the target intentionally offset with respect to the trajectory to add additional gaze rotations? This aspect of the design was quite unclear to me until I found the videos. These are much richer than the current manuscript descriptions but should not be relied upon to explain how the task was designed, especially since the rationale behind the design choices could be clearer. I imagine the authors did not record spatial position during walking (other than via the scene camera) but unless otherwise constrained people tend to walk in a straight line to complete the described tasks. This issue could be important since some of the discussion in the literature around use of optic flow has centred around whether retrieving heading from flow is necessary when steering curved paths and fixating points on your future path (rather than walking straight and looking straight ahead). Indeed the MS refers to this issue in passing in the discussion. I think it is also important since the main conclusions are that eye-movements improve the quality of the motion signal for the extraction of the direction of heading (compared to head centric coordinates). It is unclear to me that this would still be true when tracking points on the ground when travelling at speed around a bend in a road.

Minor:

Title: The title states that eye-movements improve extraction of local motion information during walking, but this seems misleading (since it's not clear what the improvement is relative to) - the reader’s natural inference may incorrectly be that the authors are comparing free gaze conditions (lots of eye-movements) versus gaze fixation (reduced numbers/types of eye-movements) in this title.

Line 165 – it's not clear what ‘3 measurements’ could entail (3 recorded frames, 3 full trajectories without any data loss? Greater precision here would be helpful.

Line 233 – ‘patter’ should be ‘pattern’

Line 296 – ‘in sync’ should probably no be abbreviated (i.e. ‘in synchrony’)

Line 315-318 – I found these statements about the results a bit hard to follow because of inconsistent labelling, e.g. ‘kept their eyes fixed in both frames of reference’ – for clarity please use consistent labels throughout. Lines 309-311 refer to: “scenes, task conditions, coordinates” but elsewhere other labels (“fixed, frames of reference”) are used.

Line 320 - ‘form’ should be ‘from’

Line 330 – ‘the reduction in noise’ – which reduction? Not clear from the context.

Line 356 – ‘broadband luminance-defined stimuli’ Could the authors clarify what they mean by this?

Line 376 – “Successful extraction of FoE”. I’m not sure the authors provide a way of determining success in extraction of the FoE. Is any value of template matching error less than 90 degrees success? Or does template matching need to be close to zero? They demonstrate improved accuracy of extraction in certain conditions but in the literature a criterion value (e.g. gauging heading to within 2 deg or less) has tended to be needed to determine whether heading is gauged with sufficient accuracy to guide an action.

Line 415 – “driving fast eye-movements”. Could you be more specific? Do you mean driving (a car) or do you mean something like optokinetic nystagmus?

Line 418 – “regular” – I think you mean smooth or flat?

Reviewer #2: This paper is about an important question - whether eye movements improve local motion information during walking. The optical situation is this. Consider a rectangular frame of reference, centered on O the nodal point of an eye and attached rigidly to the head of an observer. The frame of reference will move relative to the environment as the observer walks through the environment. There are six components of motion of the head frame of reference relative to fixed axes in the environment. These are three rotary components, and three translatory components. At any instant, if the rotary components are zero, the optic velocity field at O will be expanding radially, and the centre of expansion will correspond to the direction relative to the environment along which O is moving at that instance. In this case, the optic velocity field would afford clear information about the direction of heading of O. However, if, as will usually be the case, the rotary components of the head frame of reference are not all zero, then the optic velocity field at O will be the vector sum of the radial optic velocity field due to the translatory movement of the head frame of reference, and a rotary optic velocity field due to the rotation of the head frame of reference. In this case, the optic velocity field would afford noisy information about the direction of heading of O. The visual system could in principle deal with this problem by counter-rotating the eye in the head to cancel out the rotary movement of the head frame of reference. To what extent the eye can do this is an empirical question which I had hoped the paper would be answering. Maybe the 2DMD analytic method they used is directed in that direction, but it is unclear that it is, from the scant outline of the analytic method given in the paper, and the scant results presented, which do not properly address the problem. This is a great pity, because I for one would very much like to know how the active visual system works in such everyday activities as walking around the environment.

6. PLOS authors have the option to publish the peer review history of their article (what does this mean?). If published, this will include your full peer review and any attached files.

Reviewer #1: No

Reviewer #2: Yes: David N Lee

---

## [Author Response · Author response to Decision Letter 0]

22 Dec 2019

Please see uploaded document titles 'Response to Reviewers' as we were instructed.

---

## [Editor Report · Decision Letter 1]

14 Jan 2020

The combined effect of eye movements improve head centred local motion information during walking

PONE-D-19-26713R1

Dear Dr. Durant,

We are pleased to inform you that your manuscript has been judged scientifically suitable for publication and will be formally accepted for publication once it complies with all outstanding technical requirements.

With kind regards,

Doug Wylie, Ph.D.

Academic Editor

PLOS ONE
---

## [Editor Report · Acceptance letter]

22 Jan 2020

PONE-D-19-26713R1 

The combined effect of eye movements improve head centred local motion information during walking 

Dear Dr. Durant:

I am pleased to inform you that your manuscript has been deemed suitable for publication in PLOS ONE. Congratulations! Your manuscript is now with our production department. 

With kind regards,

on behalf of

Dr. Doug Wylie 

Academic Editor

PLOS ONE